# Incidence and Outcome of Patients with Cardiogenic Shock and Detection of Herpes Simplex Virus in the Lower Respiratory Tract

**DOI:** 10.3390/jcm11092351

**Published:** 2022-04-22

**Authors:** Clemens Scherer, Enzo Lüsebrink, Leonhard Binzenhöfer, Thomas J. Stocker, Danny Kupka, Hieu Phan Chung, Era Stambollxhiu, Ahmed Alemic, Antonia Kellnar, Simon Deseive, Konstantin Stark, Tobias Petzold, Christian Hagl, Jörg Hausleiter, Steffen Massberg, Martin Orban

**Affiliations:** 1Department of Medicine I, University Hospital, LMU Munich, 81377 Munich, Germany; clemens.scherer@med.uni-muenchen.de (C.S.); enzo.luesebrink@gmx.de (E.L.); leonhard.binzenhoefer@med.uni-muenchen.de (L.B.); thomas.stocker@med.uni-muenchen.de (T.J.S.); hieu.phan@med.uni-muenchen.de (H.P.C.); era.stambollxhiu@med.uni-muenchen.de (E.S.); ahmed.alemic@med.uni-muenchen.de (A.A.); antonia.kellnar@med.uni-muenchen.de (A.K.); simon.deseive@med.uni-muenchen.de (S.D.); konstantin.stark@med.uni-muenchen.de (K.S.); tobias.petzold@med.uni-muenchen.de (T.P.); joerg.hausleiter@med.uni-muenchen.de (J.H.); steffen.massberg@med.uni-muenchen.de (S.M.); 2DZHK (German Center for Cardiovascular Research), Partner Site Munich Heart Alliance, 81377 Munich, Germany; christian.hagl@med.uni-muenchen.de; 3Department of Medical Oncology and Hematology, University Hospital Zurich, 8091 Zurich, Switzerland; danny.kupka@gmail.com; 4Department of Cardiac Surgery, University Hospital, LMU Munich, 81377 Munich, Germany

**Keywords:** cardiogenic shock, myocardial infarction, herpes simplex virus, aciclovir, pneumonia

## Abstract

(1) Herpes simplex virus (HSV) reactivation in critically ill patients can cause infection in the lower respiratory tract, prolonging mechanical ventilation. However, the association of HSV reactivation with cardiogenic shock (CS) is unclear. As CS is often accompanied by pulmonary congestion and reduced immune system activity, the aim of our study was to determine the incidence and outcome of HSV reactivation in these patients. (2) In this retrospective, single-center study, bronchial lavage (BL) was performed on 181 out of 837 CS patients with mechanical ventilation. (3) In 44 of those patients, HSV was detected with a median time interval of 11 days since intubation. The occurrence of HSV was associated with an increase in C-reactive protein and the fraction of inspired oxygen at the time of HSV detection. Arterial hypertension, bilirubin on ICU admission, the duration of mechanical ventilation and out-of-hospital cardiac arrest were associated with HSV reactivation. (4) HSV reactivation could be detected in 24.3% of patients with CS on whom BL was performed, and its occurrence should be considered in patients with prolonged mechanical ventilation. Due to the limited current evidence, the initiation of treatment for these patients remains an individual choice. Dedicated randomized studies are necessary to investigate the efficacy of antiviral therapy.

## 1. Introduction

To this day, cardiogenic shock is a cardinal driver of mortality in patients in intensive care units (ICU). It is defined by the European Society of Cardiology (ESC) as hypotension despite adequate filling status (with signs of hypoperfusion due to primary cardiac dysfunction) and affects around 10% of patients with acute myocardial infarction [1,2]. Today, patients with cardiogenic shock exhibit a 30-day mortality rate of around 50% [3,4,5]. Patients with cardiogenic shock often require long-term intensive care and mechanical ventilation, resulting in secondary complications such as pneumonia, bleeding, stroke and acute kidney injury [6]. Especially in patients with VA-ECMO, the incidence of ventilator-associated pneumonia is high, reaching 55% in a retrospective study [7]. This is in part provoked by pulmonary congestion due to retrograde ECMO flow, insufficient LV unloading, and post-resuscitation injury and aspiration [5]. Herpes simplex virus (HSV) is known to cause infections of mucocutaneous surfaces, the central nervous system and visceral organs. After primary infection, HSV remains non-replicative in the sensory ganglia [8]. Throughout the lifespan, reactivation can be triggered by local or systemic stimuli such as UV light, tissue lesions and immune system impairment [9]. Oropharyngeal reactivation can be detected in around 20–54% of critically ill patients [9]. Furthermore, HSV can be substantiated in 64% of cases with prolonged mechanical ventilation by bronchial lavage [10]. However, the impact of antiviral therapy remains controversial. In a recent meta-analysis comprising nine studies, antiviral therapy was associated with lower hospital mortality and 30-day mortality [9].

Patients with cardiogenic shock might be prone to HSV reactivation due to their frequently long-lasting mechanical ventilation. However, the impact of HSV occurrence in these patients is unknown so far. Hence, our retrospective study aimed to analyze the incidence, clinical outcome and patient characteristics associated with HSV reactivation in cardiogenic shock.

## 2. Materials and Methods

### 2.1. Study Population

After approval by the local ethics committee (IRB number: 18-001; IRB name: “Ethikkommission bei der Medizinischen Fakultät der LMU München”), patients with cardiogenic shock who were treated in the cardiac intensive care unit (ICU) of Ludwig-Maximilians-University (LMU) from 01/2010 until 08/2021 were included in the LMUshock registry, in compliance with the Declaration of Helsinki and German data protection laws. Cardiogenic shock was defined by the ESC guidelines [1], the IABP-SHOCK II trial [11] and the CULPRIT SHOCK trial [12] as hypotension despite adequate filling status (systolic blood pressure < 90 mmHg for 30 min or catecholamines to maintain systolic blood pressure > 90 mmHg) and signs of pulmonary congestion, and at least one of the following clinical and laboratory signs of hypoperfusion: (i) Clinical: altered mental status; dizziness; cold, clammy skin and extremities; oliguria with urine output < 30 mL/h; narrow pulse pressure. (ii) Laboratory: metabolic acidosis, elevated serum lactate > 2 mmol/L and elevated creatinine due to primary cardiac dysfunction. The LMUshock registry is registered at the WHO International Clinical Trials Registry Platform (DRKS00015860). All patients with mechanical ventilation from the LMUshock registry were included in our analysis.

### 2.2. Sedation, Ventilation, Mechanical Support Devices and Bronchial Lavage

Initially, all patients were ventilated in controlled biphasic positive airway pressure mode (target tidal volume 6–8 mL/kg of predicted body weight, lung protective ventilation) and respirator settings were adjusted according to arterial blood gas analyses. Where applicable, VA-ECMO and coaxial left ventricular assist device (Impella) implantation, weaning and explantation were performed as described previously [4,5,13,14]. Bronchial lavage (BL) was performed in order to identify pathogens in eligible patients in the case of an unexplained new fever, purulent pulmonary secretions, new pulmonary infiltrates, the progression of existing infiltrates, an unexplained decrease in paO_2_/FiO_2_ or an unexplained increase in inflammatory markers and catecholamines. BL samples were analyzed in the local virology department by performing a quantitative HSV1-PCR test.

### 2.3. Statistical Analysis

The statistical analysis was performed with R (version 4.0.1, The R Foundation) following the STROBE (Strengthening the Reporting of Observational Studies in Epidemiology) statement [15]. Normally distributed continuous variables were reported as mean with standard deviation and non-normally distributed continuous variables as median with interquartile ranges. The *T*-test and Mann–Whitney U test were used to compare groups, respectively. One-way analysis of variance and the Kruskal–Wallis rank sum test was used, respectively, to compare three or more groups. Categorical variables were reported as absolute numbers and percentages, and the Chi-square or Fisher’s exact test was utilized for comparison. All tests were two-tailed and *p*-values < 0.05 were considered to be significant. To assess the correlation of clinical and laboratory parameters with HSV infection, univariate and multivariate binary logistic regression models were used. Covariates included age, male gender, hypertension, diabetes mellitus, body mass index, first measured lactate in ICU, first measured creatinine in ICU, first measured bilirubin in ICU, Horowitz (paO_2_/FiO_2_) index, cardiac arrest, out-of-hospital cardiac arrest, myocardial infarction, VA-ECMO treatment, coaxial left ventricular assist device (Impella) treatment, renal replacement therapy, tracheotomy, average systolic blood pressure and duration of mechanical ventilation. Parameters for multivariate analysis were stepwise selected by Akaike information criterion (AIC) with backward direction and 1000 bootstrap iterations using the step AIC function of the R package MASS (version 7.3–51.6).

## 3. Results

### 3.1. Study Population and Baseline Characteristics

At the time of analysis, 1200 patients were included in the LMUshock registry. Of these, 837 patients with mechanical ventilation were available for analysis. The mean age was 65 ± 15 years, with 74% being male. The median body mass index was 27 kg/m^2^. Cardiac arrest occurred in 72% of patients and out-of-hospital cardiac arrest occurred in 36% with a median cardio-pulmonary resuscitation duration of 20 min. The primary causes of cardiogenic shock were ST-elevation myocardial infarction (STEMI) in 34% of patients and non-ST-elevation myocardial infarction (NSTEMI) in 25% of patients (Table 1).

HSV reactivation was confirmed in 5.3% of the mechanical ventilation cohort. Patients were assigned into three groups: patients with HSV detection in BL (*n* = 44), patients with no HSV detection in BL (*n* = 137) and patients without BL performed (*n* = 656) (Appendix A).

### 3.2. Comparison between Patients with HSV vs. without HSV Detection

There was no statistical difference concerning the baseline characteristics between patients with HSV detection in BL and patients without HSV detection (Table 1). For patients with HSV detection in BL, the median ICU stay duration was significantly longer in patients with HSV reactivation (23 days, IQR 19–29, vs. 13 days, IQR 8–21, *p* < 0.01). Moreover, the duration of mechanical ventilation was longer in the HSV positive group with 454 h (IQR 326–604) vs. 255 h (IQR 161–458) (*p* < 0.01) in the HSV negative group. Significantly more patients with HSV detection were tracheotomized in comparison to HSV negative patients (57% vs. 35%, *p* = 0.02). Median SAPS2 score (76 vs. 74, *p* = 0.35), median SOFA score (13 vs. 13, *p* = 0.72), lactate on ICU admission (6.3 vs. 7.1 mmol/L, *p* = 0.37), glomerular filtration rate (38 vs. 45 mL/min, *p* = 0.10) and platelet count (209 vs. 204 G/L, *p* = 0.70) were not statistically different between both groups. However, bilirubin levels were significantly higher in the group with HSV detection (1.3 vs. 0.8 mg/dL, *p* < 0.01). Renal replacement therapy tended to be required more frequently in the HSV positive group (68% vs. 51%, *p* = 0.07). Usage of VA-ECMO treatment (57% vs. 58%, *p* = 0.99), coaxial left ventricular assist device (Impella) treatment (21% vs. 17%, *p* = 0.74) and IABP (0% vs. 3%, *p* = 0.58) were comparable between both groups. All ICU parameters are summarized in Table 2. Mortality rate after 30 days was 22.7% for mechanically ventilated patients with HSV detection in BL and 43.8% for patients without HSV detection in BL.

### 3.3. Characteristics of HSV Reactivation and Antiviral Therapy

The median time from intubation to first HSV detection was 11 days (IQR 9–13) (Figure 1A). The median of the highest HSV copy number per patient was 32.5 million per mL, but the HSV copy number varied widely between patients (Figure 1B). Out of 44 patients with HSV detected, 42 patients received intravenous antiviral therapy with aciclovir for a median of 7.3 days, whereas two patients did not receive any antiviral therapy at the physician’s discretion.

### 3.4. Impact of HSV Reactivation on Inflammation and Ventilation

As most patients received antiviral therapy, the impact of HSV detection in BL was investigated by comparing the inflammatory and ventilation parameters.

Concomitant pathogens detected seven days before until seven days after the first HSV occurrence were recorded to identify conflicting causes of inflammatory activity (Table 3). In 15 out of 44 patients with HSV detection, bacteria were substantiated. Coagulase-negative staphylococci were detected in 5 of these 15 patients, possibly reflecting contamination. Candida yeasts were found in 34 out of 44 patients with HSV detection. Cytomegalovirus was identified in one patient.

CRP values were significantly higher in the second and third quartile of HSV detection time points for patients with HSV, in comparison to patients without HSV detected in BL (*p* = 0.04, Figure 2A). Furthermore, CRP significantly declined after that in patients with HSV compared to patients without HSV detected in BL (*p* = 0.02 vs. *p* = 0.30, Figure 2A). Concerning mechanical ventilation parameters, the fraction of inspired oxygen increased during the second and third quartile of HSV detection time points compared to patients without HSV detected in BL (*p* < 0.01, Figure 2B). No change in creatinine values was observed between patients with and without HSV (*p* = 0.25, Figure 2C).

### 3.5. Risk Factors for HSV Occurrence

A binary regression model was developed to investigate which factors could favor the occurrence and, therefore, detection of HSV in patients with mechanical ventilation for at least seven days and BL. The results of univariate analysis are shown in Table 4. After stepwise selection of parameters, cardiovascular risk factor arterial hypertension (HR 3.240, *p* = 0.034), bilirubin measured on ICU admission (HR 2.125 per mg/dL, *p* < 0.001), the duration of mechanical ventilation (HR 1.106 per day, *p* = 0.001) and out-of-hospital cardiac arrest (HR 3.429, *p* = 0.030) were all associated with HSV occurrence (Table 4).

## 4. Discussion

This single-center study investigated the incidence and outcome of HSV detection in patients with cardiogenic shock and mechanical ventilation. The main findings of our study are as follows: (1) In 181 patients with cardiogenic shock and mechanical ventilation undergoing bronchial lavage (out of 837 with cardiogenic shock and mechanical ventilation), HSV was detected in 44 (24.3%) with a median time from intubation to HSV verification of 11 days; (2) reactivation was associated with an increase in CRP and FiO2 at the time of detection; (3) arterial hypertension, bilirubin on ICU admission, the duration of mechanical ventilation and out-of-hospital cardiac arrest were all associated with an increase in HSV occurrence; (4) the ventilation time of patients with HSV detection was longer compared to patients without HSV; and (5) the 30-days mortality rate of patients with HSV was 22.7%.

To the best of our knowledge, this is the first study analyzing HSV incidence in patients with cardiogenic shock. It is known that these patients display systemic inflammation, with a depletion of immune cells identified in the most severely affected patients throughout the course of the disease. This may facilitate the reactivation of HSV [16]. Interestingly, bilirubin was associated with HSV occurrence alongside arterial hypertension, the duration of mechanical ventilation and out-of-hospital cardiac arrest. Bilirubin levels on ICU admission are known to be linked to ARDS development and mortality from sepsis [17]. One may speculate that elevated bilirubin on ICU admission may indicate chronic liver injury due to cardiac deterioration, which in turn could impair the immune system.

Our data indicate that the possibility of HSV reactivation should be considered in cardiogenic shock patients with long-term mechanical ventilation, considering the observed increase in CRP and FiO2 values from day nine onwards. Concerning the impact of antiviral therapy, a meta-analysis was able to show a possible benefit regarding 30-day all-cause mortality in HSV patients treated with antiviral therapy [9]. However, large-scale randomized controlled trials confirming those results are missing, and it remains unclear if treatment is necessary for survival. In our patients with HSV detection, the mortality rate after 30 days was 22.7%, in comparison to 43.8% for patients without HSV detection in BL. This difference may be due to selection bias (as HSV patients had a longer median ventilation time of 18.9 days, thereby excluding patients who died before HSV reactivation was possible) or the availability of a specific treatment to target the presumed cause of long-term ventilation for HSV patients, in contrast to non-HSV patients. As almost all patients were treated with aciclovir in our study, we could not analyze a treatment effect.

Due to the low number of cases and other factors heavily influencing mortality, randomized controlled trials assessing the efficacy of antiviral treatment may be challenging to perform on patients with cardiogenic shock and HSV reactivation. First, prospective, multicenter registries should systemically screen patients with cardiogenic shock and long-term mechanical ventilation for HSV detection to quantify the incidence of HSV occurrence and its impact on different organ systems more precisely.

Our results cannot be generalized due to the single-center design, but this study is the largest analysis carried out in cardiogenic shock patients so far. Additionally, HSV incidence was not tested in all patients with cardiogenic shock, as BL was only performed in a subset of patients at the physician’s discretion and more patients could have had undetected HSV reactivation. However, physicians were required to perform a BL in case of an unexplained increase in inflammation markers in our ICU. Furthermore, our study cannot estimate the efficacy of antiviral treatment, as most patients were treated with aciclovir.

## Figures and Tables

**Figure 1 jcm-11-02351-f001:**
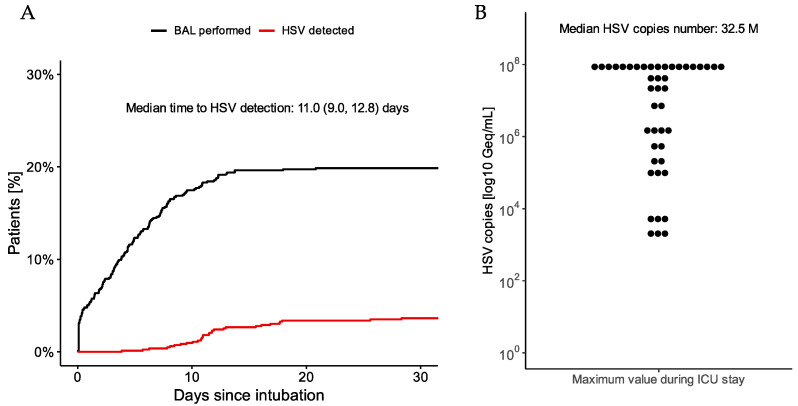
(**A**) Rates of BAL examinations (black line) and HSV occurrence (red line) since intubation for patients with cardiogenic shock. (**B**) Distribution of highest HSV copy number measured per patient during ICU stay.

**Figure 2 jcm-11-02351-f002:**
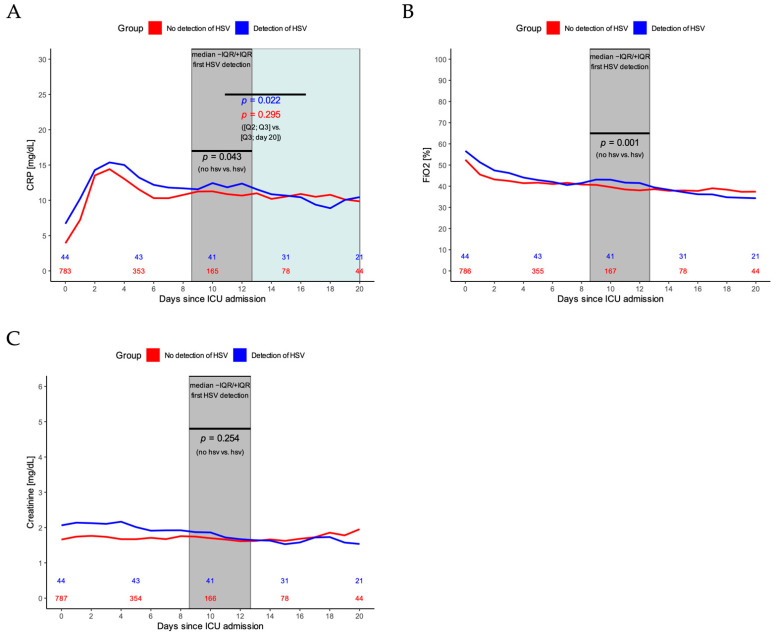
(**A**) CRP dynamics during ICU stay for patients with (blue line) and patients without HSV detection (red line). *p*-value in black indicates the statistical difference for CRP values of patients with vs. without HSV during the second and third quartile of HSV detection timepoints (grey area). *p*-values in color indicate the statistical difference for CRP values of patients with (blue line) and without HSV (red line) during the second and third quartile of first HSV detection values vs. the time after that until day 20. (**B**) Fraction of inspired oxygen (FiO2) for patients with (blue line) and without HSV detection (red line). *p*-value indicates the statistical difference for FiO2 values of patients with vs. without HSV during the second and third quartile of HSV detection timepoints (grey area). (**C**) Creatinine values for patients with (blue line) and without HSV detection (red line). *p*-value indicates the statistical difference for creatinine values of patients with vs. without HSV during the second and third quartile of HSV detection timepoints (grey area).

**Table 1 jcm-11-02351-t001:** Baseline characteristics.

Variables	All Patients *n* = 837	No BL *n* = 656	No HSV in BL *n* = 137	BL with HSV *n* = 44	*p*-Value (No HSV vs. HSV in BL)
Age, years (SD)	64.8 (14.9)	66.1 (14.9)	59.8 (14.3)	61.1 (11.5)	0.58
Male gender, *n* (%)	617 (73.7)	475 (72.4)	112 (81.8)	30 (68.2)	0.09
Body mass index, kg/m^2^ (IQR)	26.5 (24.2, 29.7)	26.3 (24.2, 29.4)	27.5 (24.8, 30.4)	25.4 (23.4, 29.7)	0.11
Previous PCI, *n* (%)	252 (30.1)	191 (29.1)	41 (29.9)	20 (45.5)	0.09
Previous CABG, *n* (%)	90 (10.8)	73 (11.1)	14 (10.2)	3 (6.8)	0.71
Previous stroke, *n* (%)	86 (10.3)	68 (10.4)	13 (9.5)	5 (11.4)	0.94
Known peripheral artery disease, *n* (%)	98 (11.7)	85 (13.0)	9 (6.6)	4 (9.1)	0.82
Smoker, *n* (%)					0.69
Active smoker	203 (24.3)	146 (22.3)	45 (32.8)	12 (27.3)	
Former smoker	152 (18.2)	125 (19.1)	19 (13.9)	8 (18.2)	
Never smoked	482 (57.6)	482 (57.6)	73 (53.3)	24 (54.5)	
Hypertension, *n* (%)	621 (74.2)	496 (75.6)	94 (68.6)	31 (70.5)	0.97
Dyslipidemia, *n* (%)	369 (44.1)	287 (43.8)	62 (45.3)	20 (45.5)	1.00
Diabetes mellitus, *n* (%)	277 (33.1)	217 (33.1)	45 (32.8)	15 (34.1)	1.00
Positive cardiovascular family history, *n* (%)	508 (60.7)	396 (60.4)	81 (59.1)	31 (70.5)	0.24
Cardiac arrest, *n* (%)	600 (71.7)	466 (71.0)	105 (76.6)	29 (65.9)	0.22
Out-of-hospital cardiac arrest, *n* (%)	302 (36.1)	249 (38.0)	40 (29.2)	13 (29.5)	1.00
Duration of cardio-pulmonary resuscitation if applicable, minutes (IQR)	20.0 (12.0, 33.0)	20.0 (12.0, 35.0)	20.0 (14.0, 30.0)	15.0 (10.0, 25.5)	0.38
Cause of cardiogenic shock, *n* (%)					0.66
Primary arrhythmia	83 (9.9)	73 (11.1)	7 (5.1)	3 (6.8)	
Decompensated CMP	95 (11.4)	75 (11.4)	13 (9.5)	7 (15.9)	
Myocarditis	23 (2.7)	15 (2.3)	5 (3.6)	3 (6.8)	
NSTEMI	205 (24.5)	163 (24.8)	35 (25.5)	7 (15.9)	
Other	83 (9.9)	64 (9.8)	14 (10.2)	5 (11.4)	
STEMI	280 (33.5)	211 (32.2)	54 (39.4)	15 (34.1)	
Valvular	68 (8.1)	55 (8.4)	9 (6.6)	4 (9.1)	
Percutaneous coronary intervention, *n* (%)	495 (59.1)	384 (58.5)	89 (65.0)	22 (50.0)	0.17

**Table 2 jcm-11-02351-t002:** ICU characteristics.

Variables	All Patients *n* = 837	No BL *n* = 656	No HSV in BL *n* = 137	BL with HSV *n* = 44	*p*-Value (no HSV vs. HSV in BL)
**Duration of ICU stay, days (IQR)**	**7.6 (2.2, 13.7)**	**5.4 (1.6, 10.3)**	**13.3 (8.4, 20.8)**	**22.8 (19.0, 28.9)**	**<0.01**
SAPS II score (IQR)	76.0 (68.0, 84.0)	76.0 (68.0, 85.0)	74.0 (66.0, 82.0)	75.5 (70.0, 83.0)	0.35
SOFA score on ICU admission (IQR)	12.0 (10.0, 14.0)	12.0 (10.0, 14.0)	13.0 (11.0, 15.0)	13.0 (11.0, 15.0)	0.72
Lactate on ICU admission, mmol/L (IQR)	6.1 (2.6, 9.5)	5.9 (2.5, 9.5)	7.1 (2.9, 9.6)	6.3 (2.6, 9.4)	0.37
pH on ICU admission (IQR)	7.3 (7.2, 7.3)	7.3 (7.2, 7.3)	7.3 (7.2, 7.3)	7.3 (7.2, 7.3)	0.70
Creatinine on ICU admission, mg/dL (IQR)	1.4 (1.1, 2.0)	1.4 (1.1, 1.9)	1.6 (1.3, 1.9)	1.7 (1.3, 2.4)	0.22
GFR on ICU admission, mL/min (IQR)	46.0 (32.5, 60.6)	47.5 (32.1, 61.8)	44.9 (34.8, 56.1)	37.8 (23.8, 55.0)	0.10
Platelet count on ICU admission, G/L (IQR)	206.0 (148.0, 259.0)	206.0 (148.0, 257.0)	204.0 (145.0, 268.0)	209.0 (155.5, 268.2)	0.70
Hemoglobin on ICU admission, g/dL (IQR)	11.6 (9.7, 13.7)	11.7 (9.8, 13.7)	11.8 (9.7, 13.9)	11.2 (9.3, 12.9)	0.60
Albumin on ICU admission, g/dL (IQR)	2.9 (2.5, 3.3)	3.0 (2.5, 3.4)	2.8 (2.4, 3.1)	2.8 (2.5, 3.2)	0.77
**Bilirubin on ICU admission, mg/dL (IQR)**	**0.9 (0.5, 1.5)**	**0.9 (0.6, 1.4)**	**0.8 (0.5, 1.4)**	**1.3 (0.9, 2.5)**	**<0.01**
Horowitz index (paO_2_/FiO_2_) on admission (IQR)	157.6 (107.0, 238.8)	170.7 (114.3, 245.8)	131.2 (86.8, 197.0)	129.2 (89.6, 202.8)	0.93
Average systolic blood pressure, mmHg (IQR)	108.6 (98.0, 119.9)	108.7 (96.5, 120.5)	106.7 (99.6, 114.8)	109.8 (102.2, 118.0)	0.09
Average diastolic blood pressure, mmHg (IQR)	57.8 (52.3, 62.5)	57.7 (51.6, 62.5)	58.4 (54.2, 63.1)	57.2 (54.4, 61.0)	0.51
Average heart rate, bpm (IQR)	84.4 (76.1, 92.9)	84.0 (75.4, 92.7)	86.7 (78.4, 93.8)	84.8 (79.0, 95.1)	0.93
Renal replacement therapy, *n* (%)	269 (32.1)	169 (25.8)	70 (51.1)	30 (68.2)	0.07
**Duration of mechanical ventilation in hours, *n* (%)**	**90.0 (19.2, 223.5)**	**57.1 (12.0, 159.1)**	**255.3 (161.0, 457.5)**	**453.8 (326.4, 604.1)**	**<0.01**
**Tracheotomy, *n* (%)**	**129 (15.4)**	**56 (8.5)**	**48 (35.0)**	**25 (56.8)**	**0.02**
Therapeutic hypothermia, *n* (%)	270 (32.3)	201 (30.6)	55 (40.1)	14 (31.8)	0.42
VA-ECMO treatment, *n* (%)	312 (37.3)	207 (31.6)	80 (58.4)	25 (56.8)	0.99
Duration VA-ECMO treatment in days, *n* (%)	4.1 (3.0)	3.3 (2.7)	5.3 (2.9)	6.1 (3.5)	0.23
SAVE score (SD)	−9.0 (5.1)	−9.0 (5.1)	−8.5 (5.1)	−10.6 (4.2)	0.13
Coaxial left ventricular assist device (Impella) treatment, *n* (%)	111 (13.3)	79 (12.0)	23 (16.8)	9 (20.5)	0.74
IABP treatment, *n* (%)	43 (5.1)	39 (5.9)	4 (2.9)	0 (0.0)	0.58
**CPC score on ICU discharge, *n* (%)**					**0.04**
**CPC 1**	**50 (6.0)**	**42 (6.4)**	**4 (2.9)**	**4 (9.1)**	
**CPC 2**	**121 (14.5)**	**109 (16.6)**	**9 (6.6)**	**3 (6.8)**	
**CPC 3**	**210 (25.1)**	**143 (21.8)**	**45 (32.8)**	**22 (50.0)**	
**CPC 4**	**85 (10.2)**	**60 (9.1)**	**23 (16.8)**	**2 (4.5)**	
**CPC 5**	**371 (44.3)**	**302 (46.0)**	**56 (40.9)**	**13 (29.5)**	

**Table 3 jcm-11-02351-t003:** Concomitant pathogens detected seven days prior until seven after first HSV occurrence.

Pathogen Name	Number of Patients (%)
Candida albicans	30 (68.2%)
Coagulase-negative staphylococci	5 (11.4%)
Enterococcus faecium	4 (9.1%)
Candida glabrata	3 (6.8%)
Klebsiella pneumoniae	2 (4.5%)
Cytomegalovirus	1 (2.3%)
Staphylococcus hominis	1 (2.3%)
Enterobacter cloacae	1 (2.3%)
Klebsiella oxytoca	1 (2.3%)
Candida tropicalis	1 (2.3%)
Hafnia alvei	1 (2.3%)

**Table 4 jcm-11-02351-t004:** Univariate and multivariate binary regression model for the prediction of HSV occurrence.

Risk Factor	Univariate Analysis			Multivariate Analysis after Feature Selection		
	Hazard Ratio	95% CI	*p* Value	Hazard Ratio	95% CI	*p* Value
Age (years)	0.998	0.969–1.027	0.868			
Male gender	0.337	0.142–0.796	0.013	0.377	0.129–1.086	0.070
**Hypertension**	**1.289**	**0.594**–**2.916**	**0.529**	**3.240**	**1.146**–**10.305**	**0.034**
Diabetes mellitus	1.232	0.557–2.662	0.599			
Body mass index (per kg/m^2^)	0.936	0.865–1.002	0.080	0.934	0.852–1.006	0.105
First lactate on ICU (per mmol/L)	0.958	0.884–1.028	0.260			
Creatinine on admission (per mL/min)	1.248	0.904–1.734	0.173			
Renal replacement therapy	1.830	0.853–4.111	0.129			
**Bilirubin on admission (per mg/dL)**	**1.558**	**1.178**–**2.192**	**0.005**	**2.125**	**1.495**–**3.216**	**<0.001**
**Duration of mechanical ventilation (per day)**	**1.063**	**1.015**–**1.116**	**0.011**	**1.106**	**1.047**–**1.176**	**0.001**
Horowitz (paO_2_/FiO_2_) index on admission (per mmHg/%)	1.001	0.997–1.006	0.590			
Average systolic blood pressure (per mmHg)	1.026	0.996–1.058	0.094	1.034	0.994–1.079	0.104
Cardiac arrest	0.654	0.293–1.493	0.304	0.336	0.105–1.031	0.059
**Out-of-hospital cardiac arrest**	**1.442**	**0.634**–**3.202**	**0.372**	**3.429**	**1.152**–**10.859**	**0.030**
Myocardial infarction	0.347	0.161–0.738	0.006			
VA-ECMO treatment	0.656	0.312–1.389	0.267			
Coaxial left ventricular assist device (Impella) treatment	1.197	0.471–2.884	0.695	2.596	0.789–8.624	0.113
Tracheotomy	1.630	0.787–3.438	0.192			

## Data Availability

The datasets used and/or analyzed during the current study are available from the corresponding author on reasonable request.

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
