# Peer review of "Incidence and Outcome of Patients with Cardiogenic Shock and Detection of Herpes Simplex Virus in the Lower Respiratory Tract"

_jcm, 2022, doi:10.3390/jcm11092351_

Round 1

Reviewer 1 Report

This is an interesting study about prevalence of HSV detection on BL of patients with cardiogenic shock. The study has limitation related to its retrospective nature, and some clarifications need to be done.

Line 65 page 2: please check punctuation (e.g.: add i) and ii) before “clinical” and “laboratory”).

Why male gender has been included in multivariate analysis?  In this model it seems that no-one of the variable considered is specifically a predictor of HSV detection. Why variables about mechanical ventilation have not been included, but only clinical variables?

Why the analysis on mortality has been performed on the whole population (including also those without BL?). It would be more interesting to analyze mortality in the two group with BL differing on HSV detection. At the same time, how the authors justify the reduced mortality of patients with HSV detection (that is in particular more than halved compare to patients without HSV detection)?

Please better specify the selection bias related to the choice to perform BL only to a small number of the entire cohort. The mechanism behind the choice or not to perform BL is not completely clear, as there is not a clear protocol to perform it, therefore I would limit the entire analysis to only patients with BL performed. Please also add if possible a table explaining reasons for BL execution.

Author Response

Reviewer #1 had 4 comments:

This is an interesting study about prevalence of HSV detection on BL of patients with cardiogenic shock. The study has limitation related to its retrospective nature, and some clarifications need to be done.

We thank the reviewer for giving us the possibility to improve our manuscript based on his/her expert recommendations.

  1. Line 65 page 2: please check punctuation (e.g.: add i) and ii) before “clinical” and “laboratory”).

We change the punctuation as recommended by the reviewer:

“i) clinical: altered mental status, dizziness, cold, clammy skin and extremities, oliguria with urine output < 30ml/h, narrow pulse pressure; ii) laboratory: metabolic acidosis, elevated serum lactate > 2mmol/l, elevated creatinine, due to primary cardiac dysfunction.”

(Methods, p. 2, line 67)

  1. Why male gender has been included in multivariate analysis? In this model it seems that no-one of the variable considered is specifically a predictor of HSV detection. Why variables about mechanical ventilation have not been included, but only clinical variables?

As recommended by reviewer #2, bilirubin on ICU admission was compared. As this parameter differed significantly between BL without HSV and BL with HSV detection, it was added to multivariate model as well as Horowitz (paO2/FiO2) index. Furthermore, only patients with BL were included in the model. After re-analysis, cardiovascular risk factor hypertension and bilirubin on ICU admission were significantly associated with HSV occurrence (see Table 4 below).

Male gender was included in the multivariate analysis because it was significantly associated with HSV infection in the univariate analysis.

Risk factor

Univariate analysis

Multivariate analysis after feature selection

Hazard Ratio

95% CI

p value

Hazard Ratio

95% CI

p value

Age [years]

0.998

0.969 - 1.027

0.868

Male gender

0.337

0.142 - 0.796

0.013

0.394

0.149 - 1.042

0.059

Hypertension

1.289

0.594 - 2.916

0.529

2.706

1.066 - 7.576

0.045

Diabetes mellitus

1.232

0.557 - 2.662

0.599

Body mass index [per kg/m2]

0.936

0.865 - 1.002

0.080

0.929

0.847 - 1.002

0.084

First lactate on ICU [per mmol/L]

0.958

0.884 - 1.028

0.260

Creatinine on admission [per mL/min]

1.248

0.904 - 1.734

0.173

Bilirubin on admission [per mg/dL]

1.558

1.178 - 2.192

0.005

1.662

1.186 - 2.481

0.007

Horowitz (paO2/FiO2) index on admission [per mmHg/%]

1.001

0.997 - 1.006

0.590

Cardiac arrest

0.654

0.293 - 1.493

0.304

Out-of-hospital cardiac arrest

1.442

0.634 - 3.202

0.372

2.285

0.911 - 5.792

0.078

Myocardial infarction

0.347

0.161 - 0.738

0.006

0.481

0.193 - 1.197

0.113

VA-ECMO treatment

0.656

0.312 - 1.389

0.267

Coaxial left ventricular assist device (Impella) treatment

1.197

0.471 - 2.884

0.695

  1. Why the analysis on mortality has been performed on the whole population (including also those without BL?). It would be more interesting to analyze mortality in the two group with BL differing on HSV detection. At the same time, how the authors justify the reduced mortality of patients with HSV detection (that is in particular more than halved compare to patients without HSV detection)?

As recommended by the reviewer, we focused the mortality analysis only on patients, for whom BL was performed. Mortality was still statistically significantly lower in BL HSV compared to the BL non-HSV group. We think this is primarily a selection bias as explained in the Discussion, which we are unsure how to compensate for:

“In our patients with HSV detection mortality rate after 30 days was 22.7% in comparison to 43.8% for patients without HSV detection in BL. This difference may be due to selection bias, as HSV patients had a longer median ventilation time of 18.9 days thereby excluding patients who died before HSV reactivation was possible, or due to the availability of a specific treatment to target the presumed cause of long-term ventilation for HSV patients in contrast to non-HSV patients.”

(Discussion, p. 9, l. 227)

  1. Please better specify the selection bias related to the choice to perform BL only to a small number of the entire cohort. The mechanism behind the choice or not to perform BL is not completely clear, as there is not a clear protocol to perform it, therefore I would limit the entire analysis to only patients with BL performed. Please also add if possible a table explaining reasons for BL execution.

The indication to perform a BL is explained in the Methods section. However, this is a retrospective study and physicians could have deviated from those factors:

“Bronchial lavage (BL) was performed in order to identify pathogens in eligible patients in case of unexplained new fever, purulent pulmonary secretions, new pulmonary infiltrates, progression of existing infiltrates, unexplained decrease of paO2/fiO2 or unexplained increase in inflammatory markers and catecholamines”

(see Methods, p. 2, l. 80).

Furthermore, we specified the possible selection bias error in the limitations segment:

“Additionally, HSV incidence was not tested in all patients with cardiogenic shock, as BL was only performed in a subset of patients at the physician's discretion and more patients could have had undetected HSV reactivation.”

(Discussion, p. 9, l. 241)

For the analysis of risk factors for HSV occurrence, only patients with BL were included (see above comment #2).

Reviewer 2 Report

Comments to the Authors

The authors conducted a retrospective, single-center study (n=837), in which HSV reactivation was evaluated in patients with cardiogenic shock. They reported that HSV was detected in 44 out of 181 (24.3%) patients in a setting of cardiogenic shock under a support of mechanical ventilation. Also, HSV reactivation was associated with an increase in CRP and FiO2 levels. Renal impairment was related to the detection of HSV. This reviewer believe that the focus of the present paper is somewhat novel and intriguing. Followings are comments.

  1. The cause-effect relationship cannot be established. Sicker patient status may result in HSV detection, while it is possible that HSV reactivation deteriorates patient conditions.
  2. Were the differences in Figure 1 and 2A statistically significant? It looks that HSV detection was associated with better outcomes. Mortality of patients with no HSV in BL (n=137) should be compared with those with HSV in BL (n=44). In addition, the authors describe that “In 837 patients with cardiogenic shock and mechanical ventilation, HSV was detected in 44 patients (5.3%)” in the first paragraph in the Discussion (Line 186-194). This sentence may be true but peculiar. Because only 181 patients underwent bronchial lavage in the present study, the mother population should be 181. Thus, for this reviewer, “In 181 patients with cardiogenic shock and mechanical ventilation undergoing bronchial lavage, HSV was detected in 44 (24.3%)” sounds reasonable.
  3. What were indications for performing bronchial lavage? For this reviewer, this point looks vague.
  4. Rational Although no significant between-group differences were found in baseline characteristics (Table 1), patient status may be different. For instance, how about hemoglobin, albumin, and bilirubin levels, and the SOFA and APACHE scores?
  5. The authors should delve into the relation between renal impairment and HSV reactivation.

Author Response

The authors conducted a retrospective, single-center study (n=837), in which HSV reactivation was evaluated in patients with cardiogenic shock. They reported that HSV was detected in 44 out of 181 (24.3%) patients in a setting of cardiogenic shock under a support of mechanical ventilation. Also, HSV reactivation was associated with an increase in CRP and FiO2 levels. Renal impairment was related to the detection of HSV. This reviewer believe that the focus of the present paper is somewhat novel and intriguing. Followings are comments.

We thank the reviewer for giving us the possibility to approve our manuscript.

  1. The cause-effect relationship cannot be established. Sicker patient status may result in HSV detection, while it is possible that HSV reactivation deteriorates patient conditions.

We concur, it’s difficult to establish a clear cause-effect relationship in this retrospective study. But this is generally true for most studies analyzing HSV reactivation. By our work, we hope to raise awareness for possible HSV reactivation in the very sick patient cohort with cardiogenic shock. With our analysis, we hope to show the impact of HSV reactivation (be it directly or indirectly) on clinical endpoints.

  1. Were the differences in Figure 1 and 2A statistically significant? It looks that HSV detection was associated with better outcomes. Mortality of patients with no HSV in BL (n=137) should be compared with those with HSV in BL (n=44). In addition, the authors describe that “In 837 patients with cardiogenic shock and mechanical ventilation, HSV was detected in 44 patients (5.3%)” in the first paragraph in the Discussion (Line 186-194). This sentence may be true but peculiar. Because only 181 patients underwent bronchial lavage in the present study, the mother population should be 181. Thus, for this reviewer, “In 181 patients with cardiogenic shock and mechanical ventilation undergoing bronchial lavage, HSV was detected in 44 (24.3%)” sounds reasonable.

As requested by reviewer 1 alike, we focused the mortality analysis only on patients, for whom BL was performed. Mortality was still significantly lower in BL HSV compared to the no BL HSV group.

Furthermore, we changed the first paragraph of the discussion according to the reviewer’s recommendation:

“In 181 patients with cardiogenic shock and mechanical ventilation undergoing bronchial lavage (out of 837 with cardiogenic shock and mechanical ventilation), HSV was detected in 44 (24.3%) with a median time from intubation to HSV verification of 11 days”

(Discussion, p. 8, l. 204)

  1. What were indications for performing bronchial lavage? For this reviewer, this point looks vague.

The indication to perform a BL is explained in the Methods section: “Bronchial lavage (BL) was performed in order to identify pathogens in eligible patients in case of unexplained new fever, purulent pulmonary secretions, new pulmonary infiltrates, progression of existing infiltrates, unexplained decrease of paO2/fiO2 or unexplained increase in inflammatory markers and catecholamines” (see Methods, p. 2, l. 80). However, this is a retrospective study and physicians could have deviated from those recommendations.

  1. Rational Although no significant between-group differences were found in baseline characteristics (Table 1), patient status may be different. For instance, how about hemoglobin, albumin, and bilirubin levels, and the SOFA and APACHE scores?

We added hemoglobin, albumin, bilirubin levels, Horowitz index (paO2/FiO2) and SOFA score on ICU admission to Table 2. Interestingly, the only parameters which differed significantly between groups was bilirubin, which we added to our model (see below comment #5)

  1. The authors should delve into the relation between renal impairment and HSV reactivation

As recommended by the reviewer, bilirubin on ICU admission was compared. As this parameter differed significantly between BL without HSV and BL with HSV detection, it was added to multivariate model as well as Horowitz (paO2/FiO2) index. Furthermore, only patients with BL were included in the model. After reanalysis, cardiovascular risk factor hypertension and bilirubin on ICU admission were significantly associated with HSV occurrence (see Table 4) and not renal impairment in the multivariate model:

Risk factor

Univariate analysis

Multivariate analysis after feature selection

Hazard Ratio

95% CI

p value

Hazard Ratio

95% CI

p value

Age [years]

0.998

0.969 - 1.027

0.868

Male gender

0.337

0.142 - 0.796

0.013

0.394

0.149 - 1.042

0.059

Hypertension

1.289

0.594 - 2.916

0.529

2.706

1.066 - 7.576

0.045

Diabetes mellitus

1.232

0.557 - 2.662

0.599

Body mass index [per kg/m2]

0.936

0.865 - 1.002

0.080

0.929

0.847 - 1.002

0.084

First lactate on ICU [per mmol/L]

0.958

0.884 - 1.028

0.260

Creatinine on admission [per mL/min]

1.248

0.904 - 1.734

0.173

Bilirubin on admission [per mg/dL]

1.558

1.178 - 2.192

0.005

1.662

1.186 - 2.481

0.007

Horowitz (paO2/FiO2) index on admission [per mmHg/%]

1.001

0.997 - 1.006

0.590

Cardiac arrest

0.654

0.293 - 1.493

0.304

Our-of-hospital cardiac arrest

1.442

0.634 - 3.202

0.372

2.285

0.911 - 5.792

0.078

Myocardial infarction

0.347

0.161 - 0.738

0.006

0.481

0.193 - 1.197

0.113

VA-ECMO treatment

0.656

0.312 - 1.389

0.267

Coaxial left ventricular assist device (Impella) treatment

1.197

0.471 - 2.884

0.695

We discussed elevated bilirubin levels in the discussion:

“Interestingly, bilirubin was associated with HSV occurrence beside cardiovascular risk factor hypertension. Bilirubin levels on ICU admission are known to be linked to ARDS development and mortality in sepsis [17]. One may speculate, elevated bilirubin on ICU admission may indicate chronic liver injury due to cardiac deterioration, which in turn could impair the immune system.”

(Discussion, p.9, l. 215)

Round 2

Reviewer 1 Report

The paper much improved after revision, however it is still not clear to me on what was based the selection of variables first for univariate analysis (e.g: why duration off ICU stay, average systolic blood pressure, renal replacement therapy, duration of mechanical ventilation, tracheotomy have not been analyzed at univariate analysis) and secondly to be included in the multivariate analysis (e.g. hypertension and out of of hospital cardiac arrest have a p>> 0.1, so why to include these variable also in the multivariate model?). Was the choice of variables arbitrary? If so explain why.

Author Response

 Reviewer #1 had 1 comment: 

The paper much improved after revision, however it is still not clear to me on what was based the selection of variables first for univariate analysis (e.g: why duration off ICU stay, average systolic blood pressure, renal replacement therapy, duration of mechanical ventilation, tracheotomy have not been analyzed at univariate analysis) and secondly to be included in the multivariate analysis (e.g. hypertension and out of of hospital cardiac arrest have a p>> 0.1, so why to include these variable also in the multivariate model?). Was the choice of variables arbitrary? If so explain why.

We thank the reviewer for giving us the possibility to improve our manuscript based on his/her expert recommendations.

We included the parameters average systolic blood pressure, renal replacement therapy, tracheotomy and duration of mechanical ventilation as suggested by the reviewer. However, we abstained from including duration of ICU stay, as it is highly dependent on duration of mechanical ventilation (interaction) and may only be of interest in retrospect.

The selection of variables for the multivariate model is not p-value but AIC based. We used the stepAIC R function as described in the Methods section. Therefore, variables with a p>0.05 are included in the model.

After re-analysis as suggested, our multivariate model changed (see below):

Risk factor

Univariate analysis

Multivariate analysis after feature selection

Hazard Ratio

95% CI

p value

Hazard Ratio

95% CI

p value

Age [years]

0.998

0.969 - 1.027

0.868

Male gender

0.337

0.142 - 0.796

0.013

0.377

0.129 - 1.086

0.070

Hypertension

1.289

0.594 - 2.916

0.529

3.240

1.146 - 10.305

0.034

Diabetes mellitus

1.232

0.557 - 2.662

0.599

Body mass index [per kg/m2]

0.936

0.865 - 1.002

0.080

0.934

0.852 - 1.006

0.105

First lactate on ICU [per mmol/L]

0.958

0.884 - 1.028

0.260

Creatinine on admission [per mL/min]

1.248

0.904 - 1.734

0.173

Renal replacement therapy

1.830

0.853 - 4.111

0.129

Bilirubin on admission [per mg/dL]

1.558

1.178 - 2.192

0.005

2.125

1.495 - 3.216

<0.001

Duration of mechanical ventilation [per day]

1.063

1.015 - 1.116

0.011

1.106

1.047 - 1.176

0.001

Horowitz (paO2/FiO2) index on admission [per mmHg/%]

1.001

0.997 - 1.006

0.590

Average systolic blood pressure [per mmHg]

1.026

0.996 - 1.058

0.094

1.034

0.994 - 1.079

0.104

Cardiac arrest

0.654

0.293 - 1.493

0.304

0.336

0.105 - 1.031

0.059

Out-of-hospital cardiac arrest

1.442

0.634 - 3.202

0.372

3.429

1.152 - 10.859

0.030

Myocardial infarction

0.347

0.161 - 0.738

0.006

VA-ECMO treatment

0.656

0.312 - 1.389

0.267

Coaxial left ventricular assist device (Impella) treatment

1.197

0.471 - 2.884

0.695

2.596

0.789 - 8.624

0.113

Tracheotomy

1.630

0.787 - 3.438

0.192
